# Entomological surveillance of mosquitoes (Diptera: Culicidae), vectors of arboviruses, in an ecotourism park in Cachoeiras de Macacu, state of Rio de Janeiro-RJ, Brazil

**Thamiris D'A. Balthazar** [1]*, **Danielle A. Maia**[2,3], **Alexandre A. Oliveira**[1], **William A. Marques**[1], **Amanda Q. Bastos**[2,3], **Mauricio L. Vilela**[1], **Jacenir R. S. Mallet**[1,4]

1 Laboratório interdisciplinar de vigilância entomológica em diptera e hemíptera, Instituto Oswaldo Cruz–IOC, Rio de Janeiro, Brazil, 2 Laboratório de diptera, Instituto Oswaldo Cruz–IOC, Rio de Janeiro, Brazil, 3 Universidade Federal Rural do Rio de Janeiro–UFRRJ, Seropédica, Brazil, 4 Universidade Iguaçu–UNIG, Nova Iguaçu, Brazil

☯ These authors contributed equally to this work.
* thamirisbalthazar@gmail.com

**Data Availability Statement:** All relevant data are within the manuscript and its Supporting Information files.

## Abstract

Arboviruses are arthropod-dependent viruses to complete their zoonotic cycle. Among the transmitting arthropods, culicids stand out, which participate in the cycle of several arboviruses that can affect humans. The present study aimed to identify species of culicidae and to point out the risk of circulation, emergency, or reemergence of pathogenic arboviruses to humans in the region of the Jequitibá headquarters of the Parque Estadual dos Três Picos (PETP), in Cachoeiras de Macacu, state of Rio de Janeiro, Brazil. Sampling was carried out at five Sample Points (SP) demarcated on trails from the headquarters, with CDC light traps, HP model with dry ice attached to the side, for 48 hours of activity each month. Additionally, active catches were made with a castro catcher in the period of one hour per day in the field, from six to eleven o'clock in the morning, in each PM. After the captures, thematic map was assembled using the ArcGIS 10 software and performing a multidimensional scaling (MDS). A total of 1151 specimens were captured and the presence of culicids already incriminated as vectors of arboviruses circulating in the region was observed: *Aedes fluviatilis* Lutz, 1904 (71 specimens); *Aedes scapularis* Rondani, 1848 (55 specimens); *Haemagogus leococelaenus* Dyar and Shannon, 1924 (29 specimens). In addition to the subgenus *Culex (culex)* spp. (163 specimens). In this sense, we highlight the importance of strengthening the actions of continuous entomological surveillance of the emergence and re-emergence of new arboviruses in ecotourism visitation parks.

## Introduction

Mosquitoes have a cosmopolitan distribution and are vectors of human and animal pathogens of global occurrence. In Brazil, they are incriminated as transmitters of four main arboviruses

**Funding:** This study was partially funded by the Conselho Nacional de Desenvolvimento Científico e Tecnológico (CNPq), funding the master's scholarship of the main author Thamiris Balthazar (Case number: 131750/2016-0) and by the Coordenação de Aperfeiçoamento de Pessoal de Nível Superior - Brazil (CAPES) - Financial Code 001. There was no additional external funding received for this study.

**Competing interests:** The authors have declared that no competing interests exist.

with mandatory notification in: dengue, zika, chikungunya and yellow fever [1–3]. Among the culicids already incriminated as transmitters of arboviruses to man, in urban and peri-urban areas, mosquitoes belonging to the genus *Aedes* stands out, in view of the direct incrimination of the species *Aedes aegypti* Linnaeus, 1762 and *Aedes albopictus* Skuse, 1894 as transmitters of these mandatory reporting arboviruses [4–7]

According to the epidemiological bulletin of arboviroses N°001 / 2018, the incidence rates of probable cases per 100 thousand inhabitants, for dengue, zika and chukungunya decreased, when comparing the years 2016 with the year 2017 in the state of Rio de Janeiro [8]. The yellow fever virus, on the other hand, showed great highlights between the years 2016 to 2017, with the expressive increase of cases and the occurrence in places where for a long time it was not notified [9, 10]. In the state of Rio de Janeiro there were 779 confirmed cases and 262 deaths for the period from July / 2016 to June / 2017 according to Epidemiological Report 084/2017 [11]. Presenting in the period from 2017 to 2018 a dispersion along the Brazilian east coast of the Atlantic Forest biome, where the characteristics of the fauna of vectors and primates favored the dispersion of the virus where there were no cases recorded for decades [12].

In nature, arboviruses maintain a complete and restricted zoonotic cycle between wild mosquitoes and wild animals [1, 2]. However, there may be human insertion in this dynamic due to exposure caused by anthropic changes in the environment, the frequency in protected areas for ecotourism activities or other profit-making activities, and even the presence of homes, providing contact with females of infected wild mosquitoes. In this contact with the wild cycles, man can become infected, becoming a source of infection for urban mosquitoes, favoring the emergence of urban cycles of these diseases [2].

Understanding the ecology of vector mosquito species is of paramount importance for later understand the dynamics of the transmission of these arboviruses to man [13]. In addition, the identification of culicidae fauna in ecotourism regions allows speculation about the possible transmission cycles that may occur in these areas or even be introduced or reintroduced, and to seek alternatives for the prevention of the emergence of these diseases to the local and visiting human community.

Thus, the present study aims to understand the distribution of the fauna of culecidae present in the Três Picos State Park (PETP), located in the municipality of Cachoeiras de Macacu, state of Rio de Janeiro. To assess the possibility of circulation of some of these arboviruses within the limits of the PETP headquarters, in addition to alerting to the possible introductions or reemergence of any new arboviruses due to the presence of their possible incriminated vector.

## Methods

The study was carried out at the Jequitibá headquarters of the Três Picos State Park, located in Cachoeiras de Macacu under the collection authorization (N° 058/2015) of the State Environmental Institute (INEA).

### i. Study area

The park consists of a fragment of Atlantic Forest, characterized by a dense rain forest and a tropical climate [14]. For the present study, five different Sample Points (SP) were demarcated within the limits of the Jequitibá headquarters, intended for the practice of ecotourism activities, being designated as: Parque Trail (SP 1): S: 22°24.834 'HO: 42°36 .825 '; Cristáis Trail (SP 2): S: 22° 24,964 'HO: 42° 36,543'; Jequitibá trail (SP 3): S: 22° 25.067 'HO: 42° 36.610'; Lookout trail (SP 4): S: 22° 25.067 'HO: 42° 36.610'; Visitor trail (SP 5): S: 22° 25.067 'HO: 42° 36.610' (Fig 1).

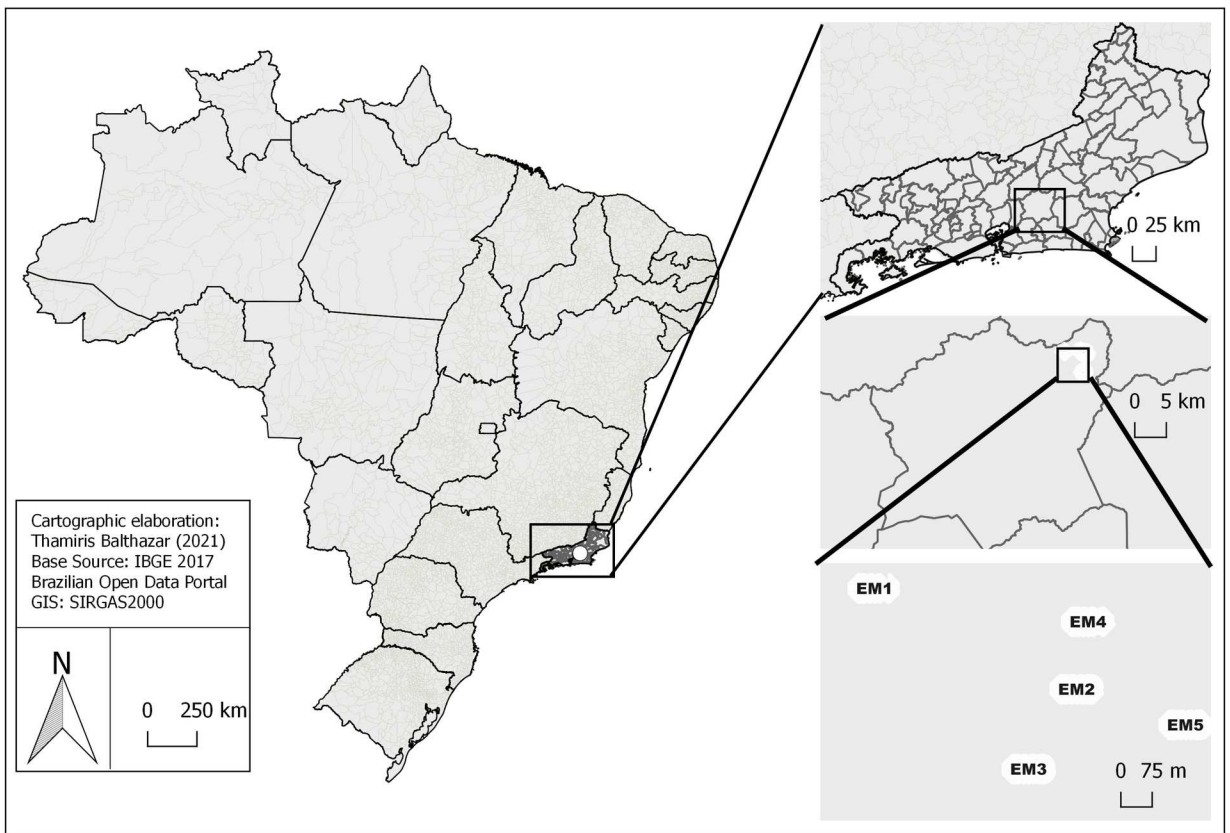

**Fig 1. Sample Points in the area of the Três Picos State Park (PETP) Cachoeiras municipality of Macacu, state of Rio De Janeiro.** Where (EM1) monitoring station 1, (EM2) monitoring station 2, (EM3) monitoring station 3, (EM4) monitoring station 4 and (EM5) monitoring station 5. Geographical meshes of the Brazilian Open Data Portal and a map created by the main author Thamiris Balthazar.

The collections were divided into two climatic periods: dry period and rainy period, following the distribution of months proposed by Minuzzi et al. [15]. In which the first encompassed the months of March to August 2017 and the rainy period covered the months of November 2016 to February 2017, together with the months of September and October 2017. Thus, eight collection campaigns were carried out, with a durability of two consecutive months, and an interval of one month between each one.

## ii. Capture methods

The specimens were captured with the aid of light traps type CDC, model HP, adapted with the addition of a styrofoam apparatus containing dry ice, attached to the side of the trap. In this way, the elimination of $CO_2$ from the apparatus worked as an additional attraction. The CDCs were installed in each of the determined sample points, at a height of 1.5 meters from the ground, and kept running in 2 cycles of 24 hours per field campaign, where at the end of each cycle the cages were changed. In addition, active captures were made with the help of a Castro catcher during the period of 1 hour at each demarcated sampling point of collection. Totaling 58 hours of sampling effort per field campaign.

## iii. Taxonomic identification

All specimens were preserved at low temperatures for proper assembly on entomological pins, identification, and storage in the entomological collection of the Diptera Laboratory, Oswaldo

Cruz Institute—IOC. And, the identification of the species was carried out through direct observation of the morphological characters evidenced by the stereomicroscope microscope (Zeiss®), and using dichotomous keys elaborated by Lane [16, 17], Faran & Linthicum [18], Consoli & Lourenço-de -Oliveira [19] and Forattini [20].

### iv. Spatial analysis

To assist in visualizing the distribution of the collected specimens, thematic maps were generated. Therefore, forest coverage rates were estimated manually using ArcGIS 10 software. A Landsat 7 satellite image (from 2002) obtained from the United States Geological Survey platform "LandsatLook" (http://landsatlook.usgs.gov/viewer.html) was used to outline the forest cover polygons. The image resolution was 30 meters for bands 1 to 7 and 15 meters for band eight, with the strips merged resulting in a final image of 15 meters in resolution. For the composition of mosquito communities it was evaluated using a multidimensional scale (MDS). MDS is a method to measure the similarity between data sets, which in this study refers to the composition of Culicidae populations (data sets) in each sampling unit [21, 22]. For the structuring and analysis of the databases, the programs Microsoft Excel and SPSS 23 were used.

## Results

At the end of the collections, a total of 1149 specimens were captured in their adult form, with 840 specimens collected in the rainy season (Table 1) and 309 specimens in the dry period (Table 2). Eight genera were identified: *Culex* (210 specimens); *Aedes* (163 specimens); *Haemagogus* (32 specimens); *Limatus* (338 specimens) *Runchomyia* (52 specimens); *Trichoprosopon* (165 specimens); *Wyeomyia* (166 specimens); *Anopheles* (17 specimens); *Uranotaenia* (6 specimens); *Sabethes* (2 specimens). Noteworthy are the most frequent species, among those considered of medical and veterinary importance: *Aedes fluviatilis* Lutz, 1904 (71 specimens); *Aedes scapularis* Rondani, 1848 (55 specimens); *Haemagogus leococelaenus* Dyar and Shannon, 1924 (29 specimens). In addition to the subgenus *Culex (culex)* spp. (163 specimens);

Two thematic maps were made, one for the rainy season (Fig 2) and another for the dry period (Fig 3) where it is possible to visualize the distribution of species of medical and veterinary interest in each MS, and these were correlated with the arboviruses of according to the literature.

Among the species captured during the study, Limatus durhamii Theobald, 1901 was present in all SP in both climatic periods, totaling 145 specimens in the rainy season and 42 specimens in the dry period. The subgenus *Culex (culex)* spp was widely distributed throughout SP, in different proportions, during the rainy season, with 159 specimens in the total period (Fig 2; Table 1), however it was more sporadic for the dry period, which was present only in SP2 (1 specimen) and SP 4 (3 specimens).

Among the species belonging to the genus *Aedes*, *Aedes terrens* Walker, 1856 was identified comprising 1.31% (11 specimens) of the culicidian fauna of the PETP headquarters in the rainy season and 2.26% (7 specimens) in the dry season. In addition, other species of this genus have also been captured, such as, *Ae. fluviatilis*, with 67 specimens in the rainy season and only 4 in the dry season; *Aedes rhyacophilus* Costa-Lima, 1933 with only 4 specimens in the rainy season and *Aedes serratus* Theobald, 1901, also present only in the rainy season with 5 specimens.

Another species of medical and epidemiological importance present in the study was *Ae. albopictus*, captured in the two SPs with the greatest anthropic changes, these being SP1 and SP5, totaling 7 specimens in the rainy season and 3 specimens in the dry season. *Aedes scapularis* showed a high density, which was identified in the Mirante trail (SP4) during the rainy

**Table 1. Total specimens collected by Sample Point (SP) during the rainy period.**

| | Rainy season | | | | | |
|---|---|---|---|---|---|---|
| | SP 1 | SP 2 | SP 3 | SP 4 | SP 5 | TOTAL |
| *Culex (carrollia)* spp. | 0 | 1 | 5 | 0 | 3 | 9 |
| *Culex (culex)* spp. | 13 | 2 | 40 | 68 | 36 | 159 |
| *Culex (Lutzia)* spp. | 10 | 3 | 4 | 1 | 5 | 23 |
| *Culex (melanoconion)* spp. | 0 | 2 | 2 | 0 | 0 | 4 |
| *Culex (microculex)* spp. | 1 | 0 | 0 | 2 | 6 | 9 |
| *Aedes (Stegomyia) albopictus* | 6 | 0 | 0 | 0 | 1 | 7 |
| *Aedes (Georgecraigius) fluviatilis* | 1 | 2 | 15 | 26 | 23 | 67 |
| *Aedes (Ochlerotatus) rhyacophilus* | 3 | 0 | 0 | 1 | 0 | 4 |
| *Aedes (Ochlerotatus) scapularis* | 1 | 5 | 1 | 43 | 4 | 54 |
| *Aedes (Ochlerotatus) serratus* | 3 | 1 | 0 | 1 | 0 | 5 |
| *Aedes (Protomacleaya) terrens* | 0 | 2 | 4 | 2 | 3 | 11 |
| *Haemagogus janthinomys/capricornii* | 0 | 0 | 0 | 1 | 2 | 3 |
| *Haemagogus leucocelaenus* | 1 | 1 | 5 | 13 | 6 | 26 |
| *Sabethes (sabethes) chloropterus* | 0 | 0 | 1 | 0 | 0 | 1 |
| *Sabethes (sabethes) belisarioi* | 0 | 0 | 1 | 0 | 0 | 1 |
| *Limatus durhamii* | 25 | 38 | 16 | 41 | 25 | 145 |
| *Limatus flavisetosus* | 9 | 10 | 5 | 10 | 8 | 42 |
| *Limatus paraensis* | 2 | 10 | 1 | 11 | 7 | 31 |
| *Runchomyia frontosus* | 2 | 0 | 2 | 5 | 5 | 14 |
| *Runchomyia lunatus* | 0 | 0 | 10 | 2 | 4 | 16 |
| *Runchomyia reversus* | 0 | 1 | 0 | 3 | 0 | 4 |
| *Trichoprosopon pallidiventer* | 0 | 0 | 1 | 0 | 1 | 2 |
| *Trichoprosopon digitatum digitatum* | 0 | 3 | 2 | 9 | 6 | 20 |
| *Trichoprosopon digitatum townsendi* | 0 | 0 | 0 | 0 | 0 | 0 |
| *Trichoprosopon obscurum* | 2 | 3 | 3 | 2 | 3 | 13 |
| *Trichoprosopon soaresi* | 4 | 12 | 15 | 18 | 18 | 67 |
| *Uranotaenia calosomata* | 1 | 0 | 0 | 0 | 4 | 5 |
| *Wyeomyia (phoniomyia) davisi* | 0 | 0 | 0 | 0 | 0 | 0 |
| *Wyeomyia aporonoma* | 1 | 0 | 0 | 3 | 5 | 9 |
| *Wyeomyia argenteorostris* | 0 | 1 | 13 | 5 | 8 | 27 |
| *Wyeomyia celaenocephala* | 0 | 0 | 0 | 0 | 1 | 1 |
| *Wyeomyia confusa* | 0 | 0 | 1 | 1 | 0 | 2 |
| *Wyeomyia flavifascies* | 1 | 0 | 0 | 2 | 0 | 3 |
| *Wyeomyia negrensis* | 2 | 1 | 16 | 9 | 13 | 41 |
| *Wyeomyia serratoria* | 0 | 0 | 0 | 1 | 0 | 1 |
| *Wyeomyia sp.* | 0 | 1 | 0 | 0 | 0 | 1 |
| *Anopheles (kerteszia) cruzii* | 0 | 4 | 4 | 0 | 7 | 15 |
| **Total:** | **88** | **103** | **165** | **280** | **204** | **840** |

season, 43 captured adults. This species was present in this rainy period, albeit in a low proportion, in the SP2 (Cristáis Trail) with 5 captured specimens, the visitor's trail (SP5) with 4 captured specimens, the Parque Trail (SP1) with only 1 specimen and the Jequitibá Trail (SP3) also with only one specimen collected at the site, throughout the rainy season. During the dry period, *Ae. scapularis* was captured only in SP4 (Lookout Trail).

We can also highlight two important species incriminated as vectors, *Sabethes chloropterus* Von Humboldt, 1819 and *Sabethes belisarioi* Neiva, 1908 collected in SP3 (Jequitibá Trail)

**Table 2. Total specimens collected by Sample Point (SP) during the dry period.**

| | Dry season | | | | | |
|---|---|---|---|---|---|---|
| | SP 1 | SP 2 | SP 3 | SP 4 | SP 5 | TOTAL |
| *Culex (carrollia)* spp. | 0 | 0 | 0 | 1 | 0 | 1 |
| *Culex (culex)* spp. | 0 | 1 | 0 | 3 | 0 | 4 |
| *Culex (Lutzia)* spp. | 0 | 0 | 0 | 1 | 0 | 1 |
| *Culex (melanoconion)* spp. | 0 | 0 | 0 | 0 | 0 | 0 |
| *Culex (microculex)* spp. | 0 | 0 | 0 | 0 | 0 | 0 |
| *Aedes (Stegomyia) albopictus* | 2 | 0 | 0 | 0 | 1 | 3 |
| *Aedes (Georgecraigius)fluviatilis* | 0 | 1 | 1 | 2 | 0 | 4 |
| *Aedes (Ochlerotatus) rhyacophilus* | 0 | 0 | 0 | 0 | 0 | 0 |
| *Aedes (Ochlerotatus) scapularis* | 0 | 0 | 0 | 1 | 0 | 1 |
| *Aedes (Ochlerotatus) serratus* | 0 | 0 | 0 | 0 | 0 | 0 |
| *Aedes (Protomacleaya) terrens* | 0 | 0 | 4 | 3 | 0 | 7 |
| *Haemagogus janthinomys/capricornii* | 0 | 0 | 0 | 0 | 0 | 0 |
| *Haemagogus leucocelaenus* | 0 | 0 | 0 | 3 | 0 | 3 |
| *Sabethes (sabethes) chloropterus* | 0 | 0 | 0 | 0 | 0 | 0 |
| *Sabethes (sabethes) belisarioi* | 0 | 0 | 0 | 0 | 0 | 0 |
| *Limatus durhamii* | 19 | 7 | 10 | 4 | 2 | 42 |
| *Limatus flavisetosus* | 0 | 0 | 10 | 3 | 0 | 13 |
| *Limatus paraensis* | 13 | 4 | 40 | 6 | 2 | 65 |
| *Runchomyia frontosus* | 0 | 0 | 1 | 2 | 1 | 4 |
| *Runchomyia lunatus* | 0 | 4 | 2 | 1 | 0 | 7 |
| *Runchomyia reversus* | 0 | 0 | 0 | 6 | 1 | 7 |
| *Trichoprosopon pallidiventer* | 0 | 0 | 1 | 0 | 0 | 1 |
| *Trichoprosopon digitatum digitatum* | 0 | 0 | 4 | 7 | 2 | 13 |
| *Trichoprosopon digitatum townsendi* | 0 | 0 | 1 | 0 | 0 | 1 |
| *Trichoprosopon obscurum* | 0 | 0 | 4 | 1 | 0 | 5 |
| *Trichoprosopon soaresi* | 1 | 1 | 28 | 9 | 4 | 43 |
| *Uranotaenia calosomata* | 1 | 0 | 0 | 0 | 0 | 1 |
| *Wyeomyia (phoniomyia) davisi* | 0 | 3 | 0 | 10 | 0 | 13 |
| *Wyeomyia aporonoma* | 0 | 1 | 4 | 1 | 0 | 6 |
| *Wyeomyia argenteorostris* | 1 | 0 | 12 | 2 | 2 | 17 |
| *Wyeomyia celaenocephala* | 0 | 0 | 0 | 0 | 0 | 0 |
| *Wyeomyia confusa* | 1 | 0 | 10 | 2 | 0 | 13 |
| *Wyeomyia flavifascies* | 0 | 0 | 5 | 0 | 0 | 5 |
| *Wyeomyia negrensis* | 0 | 1 | 14 | 4 | 3 | 22 |
| *Wyeomyia serratoria* | 0 | 0 | 2 | 1 | 2 | 5 |
| *Wyeomyia sp.* | 0 | 0 | 0 | 0 | 0 | 0 |
| *Anopheles (kerteszia) cruzii* | 0 | 0 | 2 | 0 | 0 | 2 |
| **Total:** | **38** | **23** | **155** | **73** | **20** | **309** |

during the rainy season, where only one specimen was collected decade. In addition to species of the *Haemagogus* genus, such as *Haemagogus leucocelaenus* Dyar and Shannon, 1924, which showed a density of 2.5% (29 specimens) of the total specimens collected in their adult form, in which 26 specimens of this species were captured during the period rainy season, and 3 caught in the dry season. And the species *Haemagogus janthinomys* Dyar, 1921, with only 3 specimens captured in the rainy season.

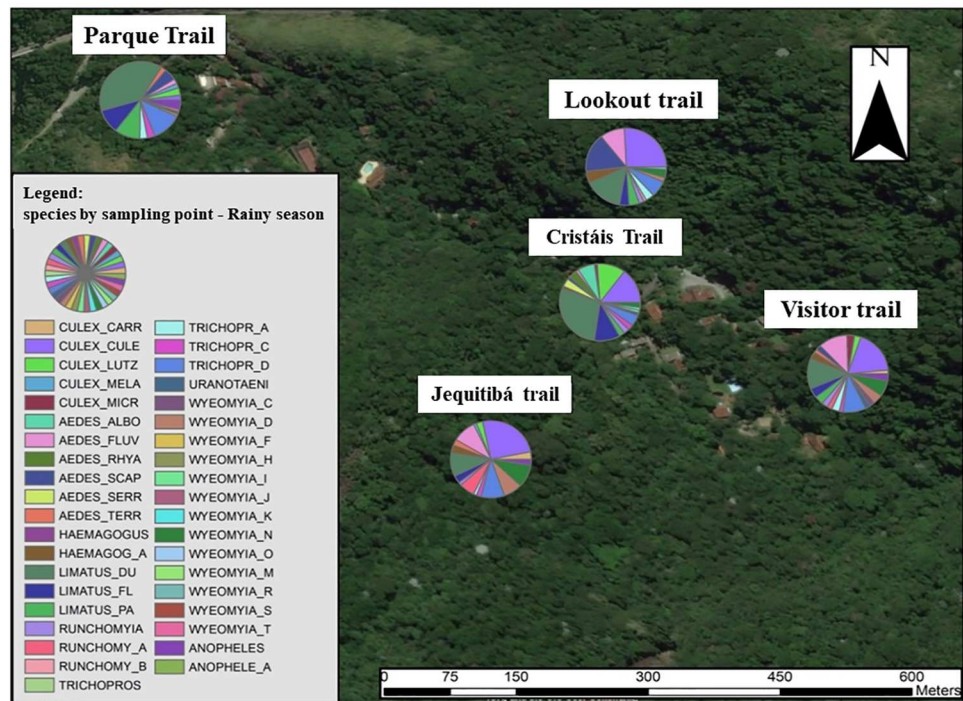

**Fig 2. Thematic map with the distribution of total species collected at each Sample Point during the rainy period.**
Landsat 7 satellite image (from 2002) obtained from the United States Geological Survey platform "LandsatLook".

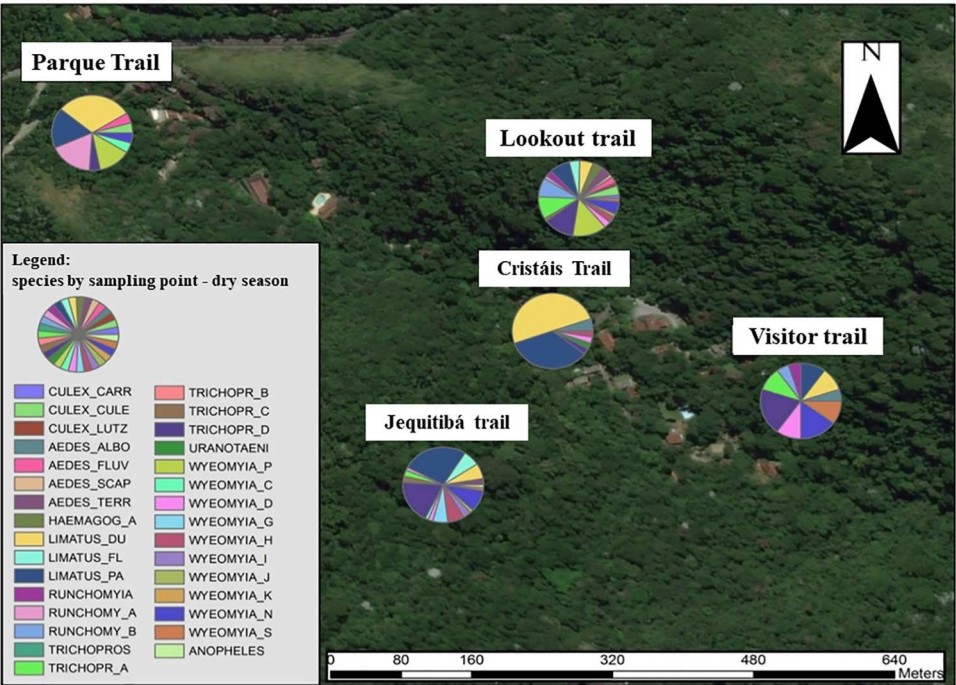

**Fig 3. Thematic map with the distribution of total species collected at each Sample Point during the dry period.**
Landsat 7 satellite image (from 2002) obtained from the United States Geological Survey platform "LandsatLook".

## Discussion

In a recent study, Abreu et al. (2019) confirmed the incrimination of the species *Haemagogus janthinomys* and *Haemagogus leucocelaenus* in the recent cases of yellow fever that occurred in the reemergence of the disease in the Atlantic Forest regions in southeastern Brazil. In this same study, the species *Sabethes chloropterus* and *Aedes scapularis* were incriminated as secondary vectors or local primary vectors of yellow fever [23]. In the present study, we point out the importance of constant entomological surveillance in areas of natural protection, such as PETP, which have ecotourism activities, considering that we observe the presence of the species *Haemagogus leucocelaenus* and *Haemagogus janthinomys*, in addition to the capture of a *Sabethes chloropterus* and *Sabethes belisarioi*.

*Aedes scapularis* was found with significant density on the trails, in the two collection periods. According to the literature, among the arboviruses in which this vector is incriminated, Forattini et al. [24–26] pointed out in their studies a strong correlation between this mosquito and the Rócio virus, a Flavivirus of great importance for the risk of reemergence in Brazil [27], in addition to its association with the Venezuelan equine encephalitis virus [28, 29].

Among the species of the genus *Aedes*, we highlight *Ae. terrens*, which has great importance in the transmission of arboviroses circulating in Brazil, with emphasis on its vectorial capacity for the Chikungunya virus, for example [30, 31]. *Ae. fluviatilis*, *Ae. rhyacophilus* and *Ae. serratus* that have also been observed, have already been incriminated by other authors as vectors or possible vectors of most arboviruses of the Togaviridae family and Flaviviridae family, in addition to some arboviruses belonging to the Bunyaviridae, Reoviridae and Rhabdoviridae families [32].

The study records the presence of *Ae. albopictus*, which is already known for its ability to be incorporated into urban and peri-urban environments, which are increasingly present, being this species of great vector importance incriminated for viruses already circulating around the park, such as Zika viruses, Dengue viruses and viruses. Chikungunya [4, 5]

In view of this, the presence of these vectors warns of the possible risk of circulation of certain arboviruses within the limits of PETP, since in the municipality of Cachoeiras de Macacu confirmed cases of arboviruses with compulsory notification in Brazil, as stated in the Epidemiological Report 045/2017 (SUS, 2017). From January to November 2017, the period in which the present study was being carried out, in parallel to these epidemiological surveys, the municipality of Cachoeira de Macacu presented 1 case of yellow fever notified on 6/2/2017, according to the epidemiological report 045/2017 [33], where species of the subgenus *Sabethes (Sabethes)* spp. captured in September, three months after this data. These factors point to the ability of this arbovirus to circulate in the PETP area, since the vector and its respective viral agent are present in this region.

Other arboviruses were also confirmed to be circulating in the municipality of Cachoeiras de Macacu, such as the Dengue virus, with an incidence of 5.4 cases per 1000 inhabitants, only two of which were confirmed; and chikungunya virus with an incidence of 1.8 cases per 1000 inhabitants, but without any confirmation [33]. The presence of the circulation of the Zika virus in the municipality was not observed, since, according to the epidemiological bulletin [33] in this period, the incidence of cases presented was zero. However, the presence of the vector and the presence of cases in neighboring municipalities warn of possible underreported circulation.

*Limatus duhramii* was found more frequently and in all SP, indicating a risk of introducing the Orthobunyavirus virus, since it has been incriminated in previous studies as a species of suspected transmission of this virus [34]. However, this evidence would require further studies on the vectorial capacity of this species.

In addition, the presence of species of the subgenus *Culex (culex)* spp. also, points to the possibility of introducing other arboviruses, since this subgenus encompasses important species of mosquitoes already incriminated as vectors of several arboviruses in the world, such as Japanese Encephalitis in Southeast Asia [35], West Nile Virus in South Africa [36] and viruses of the family Bunyaviridae already isolated [37]. Within this subgenus is *Culex (Culex) quinquefasciatus* Say, 1923, which is described with evidence and vector capacity for the Zika [38, 39]. However, a study carried out with mosquitoes of this species in the city of Rio de Janeiro did not show the same transmission capacity as Zika virus and proved that Zika virus does not replicate in mosquitoes of the species *Cx (Cux) quinquefasciatus* [40, 41]. Thus, these studies refute the hypothesis that this species is a vector of Zika virus.

The presence of incriminated or suspicious vectors in the transmission of arboviruses that have not yet been reported or considered to have been eradicated in Brazil, raises hypotheses of possible introduction, or reintroduction of these viruses, through the frequency of visitors from different regions of the world in areas of natural protection, as observed at PETP. In addition, with the notification of arboviruses already circulating in the municipality of study and the meeting of the respective vector species, they alert to the fact that PETP is a place of possible active circulation of mandatory notification arboviruses: dengue, zika, chikungunya and yellow fever.

Thus, the importance of strengthening entomological surveillance actions is highlighted. And, in addition, it is important to carry out health education actions to teach preventive measures to the local population and visitors to the PETP, aiming at blocking the circulation of arboviruses already present in the local human population, in addition to avoiding the insertion of new arboviruses in this population. Being the ecotourism visitation parks characterized as important regions for the continuous surveillance for the emergence and reemergence of new arboviruses.

## Supporting information

**S1 File.**
(XLSX)

**S2 File.**
(XLSX)

## Acknowledgments

We thank Dr. Jerônimo Alencar for his help and contribution throughout the development of this research.

## Author Contributions

**Conceptualization:** Thamiris D'A. Balthazar, Mauricio L. Vilela, Jacenir R. S. Mallet.

**Data curation:** Danielle A. Maia, Amanda Q. Bastos.

**Formal analysis:** Thamiris D'A. Balthazar, Alexandre A. Oliveira, William A. Marques, Mauricio L. Vilela, Jacenir R. S. Mallet.

**Funding acquisition:** Danielle A. Maia, Mauricio L. Vilela, Jacenir R. S. Mallet.

**Investigation:** Thamiris D'A. Balthazar, Danielle A. Maia, Jacenir R. S. Mallet.

**Methodology:** Thamiris D'A. Balthazar, Alexandre A. Oliveira, William A. Marques, Amanda Q. Bastos, Mauricio L. Vilela, Jacenir R. S. Mallet.

**Project administration:** Thamiris D'A. Balthazar, Mauricio L. Vilela, Jacenir R. S. Mallet.

**Resources:** Thamiris D'A. Balthazar.

**Supervision:** Mauricio L. Vilela, Jacenir R. S. Mallet.

**Validation:** Thamiris D'A. Balthazar, Mauricio L. Vilela, Jacenir R. S. Mallet.

**Visualization:** Mauricio L. Vilela.

**Writing – original draft:** Thamiris D'A. Balthazar.

**Writing – review & editing:** Thamiris D'A. Balthazar, Danielle A. Maia, Alexandre A. Oliveira, William A. Marques, Amanda Q. Bastos, Mauricio L. Vilela, Jacenir R. S. Mallet.

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
