## [Decision Letter · Decision Letter 0]

23 Jun 2021

PONE-D-21-16082

Entomological surveillance of mosquitoes (Diptera: Culicidae), vectors of arboviruses, in an ecotourism park in Cachoeiras de Macacu, state of Rio de Janeiro-RJ, Brazil.

PLOS ONE

Dear Dr. Balthazar,

Thank you for submitting your manuscript to PLOS ONE. After careful consideration, we feel that it has merit but does not fully meet PLOS ONE’s publication criteria as it currently stands. Therefore, we invite you to submit a revised version of the manuscript that addresses the points raised during the review process.

To comment about virological studies of collected mosquitoes.

To modify some conclusions about the circulation of arbovirus.

Consider some grammatical suggestions.

We look forward to receiving your revised manuscript.

Kind regards,

Humberto Lanz-Mendoza

Academic Editor

PLOS ONE

Journal Requirements:

'This study was financed in part by the Coordenação de Aperfeiçoamento de Pessoal de Nível Superior – Brasil (CAPES) – Finance Code 001. https://www.gov.br/capes/pt-br'

a. Please provide an amended statement that declares *all* the funding or sources of support (whether external or internal to your organization) received during this study, as detailed online in our guide for authors at http://journals.plos.org/plosone/s/submit-now

Please also include the statement “There was no additional external funding received for this study.” in your updated Funding Statement.

4. We note that Figures 1, 2 and 3 in your submission contain map and satellite images which may be copyrighted.

a. You may seek permission from the original copyright holder of Figures 1, 2 and 3 to publish the content specifically under the CC BY 4.0 license. 

5. Please upload a copy of Figure 4, to which you refer in your text. If the figure is no longer to be included as part of the submission please remove all reference to it within the text.

6. Please include a caption for figure 2.

Reviewers' comments:

Reviewer's Responses to Questions

**Comments to the Author**

1. Is the manuscript technically sound, and do the data support the conclusions?

Reviewer #1: Yes

Reviewer #2: Yes

2. Has the statistical analysis been performed appropriately and rigorously? 

Reviewer #1: Yes

Reviewer #2: N/A

3. Have the authors made all data underlying the findings in their manuscript fully available?

Reviewer #1: Yes

Reviewer #2: Yes

4. Is the manuscript presented in an intelligible fashion and written in standard English?

Reviewer #1: Yes

Reviewer #2: Yes

5. Review Comments to the Author

Reviewer #1: overall it is a simple but useful paper dealing with field research and interesting results on Yellow Fever and other important arboviruses. Since data were collected in Brazil, specifically in an eco-touristic park, it is always a surveillance monitor or geographical point of their movement or seasonal activity. Individually, most of culicidae species found are importants, but to find all of them in a small geographical area is really scary for public health reasons!! I would realy like to have lab virological study of the collected mosquitoes. I would not be surpraised findings of Yellow Fever, Dengue or other lethal pathogens!

Reviewer #2: The genesis of this manuscript is to emphasis the importance of entomological surveillance of mosquitoes in sylvatic habitats with high incursion of humans such as Natural Parks located in Atlantic Forest. This manuscript does an excellent job demonstrating significant of mosquito species surveillance in this sylvatic habitat.

The title and abstract are appropriate for the content of the text. Furthermore, the article is well constructed, the surveillances were well conducted, and analysis was well performed. However strong conclusions such as “favor the circulation of different arbovirus” (line 37 and 38 in the abstract) are not warranted, since no direct viral surveillance was carried out in the study were presented.

Thus I suggested some minor alterations to the manuscript:

Line 37 “favor the circulation of different arbovirus” This conclusion is not warranted by the study since no viral surveillance as performed. Please amend the conclusions to scope of the results presented.

Line 129 “All specimens were sacrificed”, I think that the word preserved will be more appropriated than sacrificed.

Line 269 Culex quinquefasciatus and Zika virus transmission, the authors should also mention the others published studies suggesting that Culex quinquefasciatus do not transmit Zika virus.

6. PLOS authors have the option to publish the peer review history of their article (what does this mean?). If published, this will include your full peer review and any attached files.

Reviewer #1: No

Reviewer #2: No

---

## [Author Response · Author response to Decision Letter 0]

9 Oct 2021

Figures 1, 2 and 3 were produced by lead author Thamiris Balthazar, using layers of open and public databases: Brazilian Portal for Open Data (https://dados.gov.br/dataset/malha-geometrica-dos-municipios -Brazilians) and the United States Geological Survey's "LandsatLook" platform (http://landsatlook.usgs.gov/viewer.html).

There was a change in the order of the figures. Thus, the legend named figure 3 referred to figure 2, and the legend in figure 4 referred to figure 3. Therefore, there is no fourth figure to be inserted in this article.

In line 37, the conclusion was changed, pointing to the importance of entomological surveillance in ecotourism parks.

On line 133, the word "sacrificed" has been replaced by the word "preserved".

In line 279, a continuation of the paragraph was included pointing to the articles that refute the vectorial capacity of Culex quinquefasciatus in Zika virus transmission.

---

## [Decision Letter · Decision Letter 1]

29 Nov 2021

Entomological surveillance of mosquitoes (Diptera: Culicidae), vectors of arboviruses, in an ecotourism park in Cachoeiras de Macacu, state of Rio de Janeiro-RJ, Brazil.

PONE-D-21-16082R1

Dear Dr. Balthazar,

We’re pleased to inform you that your manuscript has been judged scientifically suitable for publication and will be formally accepted for publication once it meets all outstanding technical requirements.

Kind regards,

Humberto Lanz-Mendoza

Academic Editor

PLOS ONE

Additional Editor Comments (optional):

Reviewers' comments:

Reviewer's Responses to Questions

**Comments to the Author**

1. If the authors have adequately addressed your comments raised in a previous round of review and you feel that this manuscript is now acceptable for publication, you may indicate that here to bypass the “Comments to the Author” section, enter your conflict of interest statement in the “Confidential to Editor” section, and submit your "Accept" recommendation.

Reviewer #1: All comments have been addressed

Reviewer #2: All comments have been addressed

2. Is the manuscript technically sound, and do the data support the conclusions?

Reviewer #1: Yes

Reviewer #2: Yes

3. Has the statistical analysis been performed appropriately and rigorously? 

Reviewer #1: Yes

Reviewer #2: Yes

4. Have the authors made all data underlying the findings in their manuscript fully available?

Reviewer #1: Yes

Reviewer #2: Yes

5. Is the manuscript presented in an intelligible fashion and written in standard English?

Reviewer #1: Yes

Reviewer #2: Yes

6. Review Comments to the Author

Reviewer #1: It is very amazing the amount of mosquito species found during the sampling period. Just with the reports of Sabethes species is enough to alert park visitors to have Yellow fever vaccine. Great Culicidae taxonomy work!

Reviewer #2: Previously the reviewers presented some suggestions to the manuscript. In my opinion, I observed that the issues raised were addressed in this new version. Concerning the minor comments, the authors fully addressed all points raised by the reviewers.

In conclusion I recommend the publication of the presented review article.

7. PLOS authors have the option to publish the peer review history of their article (what does this mean?). If published, this will include your full peer review and any attached files.

Reviewer #1: No

Reviewer #2: **Yes: **Alvaro Gil Araujo Ferreira

---

## [Editor Report · Acceptance letter]

15 Dec 2021

PONE-D-21-16082R1 

Entomological surveillance of mosquitoes (Diptera: Culicidae), vectors of arboviruses, in an ecotourism park in Cachoeiras de Macacu, state of Rio de Janeiro-RJ, Brazil. 

Dear Dr. Balthazar:

I'm pleased to inform you that your manuscript has been deemed suitable for publication in PLOS ONE. Congratulations! Your manuscript is now with our production department. 

Kind regards, 

on behalf of

Dr. Humberto Lanz-Mendoza 

Academic Editor

PLOS ONE